# Stroke and TIA Survivors’ Perceptions of the COVID-19 Vaccine and Influences on Its Uptake: Cross Sectional Survey

**DOI:** 10.3390/ijerph192113861

**Published:** 2022-10-25

**Authors:** Grace M. Turner, Neil Heron, Jennifer Crow, Eirini Kontou, Sally Hughes

**Affiliations:** 1Institute of Applied Health Research, University of Birmingham, Birmingham B15 2TT, UK; 2Centre for Public Health, School of Medicine, Dentistry and Biomedical Sciences, Queens University Belfast, Belfast BT12 6BA, UK; 3School of Medicine, Keele University, Staffordshire ST5 5BG, UK; 4Imperial College Healthcare NHS Trust, London W6 8RF, UK; 5Institute of Mental Health, Nottinghamshire Healthcare NHS Foundation Trust, Nottingham NG3 6AA, UK; 6School of Medicine, University of Nottingham, Nottingham NG7 2UH, UK

**Keywords:** COVID-19, vaccine, vaccine hesitancy, stroke, transient ischaemic attack, survey

## Abstract

Background: People who have experienced a stroke or transient ischaemic attack (TIA) have greater risks of complications from COVID-19. Therefore, vaccine uptake in this vulnerable population is important. To prevent vaccine hesitancy and maximise compliance, we need to better understand individuals’ views on the vaccine. Objectives: We aimed to explore perspectives of the COVID-19 vaccine and influences on its uptake from people who have experienced a stroke or TIA. Method: A cross-sectional, electronic survey comprising multiple choice and free text questions. Convenience sampling was used to recruit people who have experienced a stroke/TIA in the UK/Ireland. Results: The survey was completed by 377 stroke/TIA survivors. 87% (328/377) had either received the first vaccine dose or were booked to have it. The vaccine was declined by 2% (7/377) and 3% (11/377) had been offered the vaccine but not yet taken it up. 8% (30/377) had not been offered the vaccine despite being eligible. Some people expressed concerns around the safety of the vaccine (particularly risk of blood clots and stroke) and some were hesitant to have the second vaccine. Societal and personal benefits were motivations for vaccine uptake. There was uncertainty and lack of information about risk of COVID-19 related complications specifically for people who have experienced a stroke or TIA. Conclusion: Despite high uptake of the first vaccine, some people with stroke and TIA have legitimate concerns and information needs that should be addressed. Our findings can be used to identify targets for behaviour change to improve vaccine uptake specific to stroke/TIA patients.

## 1. Introduction

COVID-19 has had a significant impact on society since December 2019, particularly in terms of mental health, [1] physical health [2,3] and widening social inequalities [4]. One major part of the management strategy for this virus is COVID-19 vaccinations [5] For the vaccine to be effective, we need to ensure that the population is generally compliant with the vaccination programme. To address vaccine hesitancy and maximise compliance, we need to better understand individuals’ views on the vaccine. ‘Vaccine hesitancy’ refers to delay in acceptance or refusal of vaccination despite availability of vaccination services [6] and has been identified as one of the top 10 threats to global health in 2019 [7].

COVID-19 disproportionately affects certain groups, who are considered higher risk, including older people, and those with underlying medical problems like cardiovascular disease, diabetes, chronic respiratory disease, and cancer [8] Therefore, it is particularly important for these vulnerable groups to have high uptake of the vaccine and for any ‘vaccine hesitancy’ to be addressed. Previous studies have explored the views of the COVID-19 vaccine from the general population [9,10,11,12] and specific groups, including healthcare workers’ [13,14,15,16] and parents/guardians [17] However, to date, no published studies have examined the COVID-19 vaccination beliefs, behaviours, and intentions among persons who had a stroke or TIA, and who are thus at higher risk for severe COVID-19 [18] Comparably, influenza infection is associated with an increased risk of cardiovascular events for people with cardiovascular disease; however, despite well-established recommendations, only 67% of people with cardiovascular disease in the United States receive the influenza vaccine [19].

People who have experienced a stroke or TIA (Box 1) are at greater risk of serious complications of COVID-19, including pneumonia, hospitalisation and death [20,21]. There have been initiatives from stroke organisations, including the Stroke Association and British Association of Stroke Physicians, to increase vaccine uptake in this population [22,23]. However, anecdotal evidence from stroke/TIA forums, patient partners and stroke charities suggest some people who have experienced a stroke/TIA have concerns about the vaccine, including safety and effects on stroke prevention medication.

Identifying and understanding influences on individuals’ decisions to receive the COVID-19 vaccine is integral to inform strategies to improve vaccine uptake. Therefore, we conducted a survey to better understand stroke and TIA survivors’ perspectives of the COVID-19 vaccine and influences on uptake of the vaccine. As vaccine uptake is a behaviour, the survey was underpinned by a framework for understanding behaviour, COM-B (Capability, Opportunity, Motivation—Behaviour) [24].

Box 1Summary of stroke types, treatments and outcomes [25,26].*Stroke types* 
-Ischaemic stroke: caused by a blockage cutting off the blood supply to the brain. This is the most common type of stroke.-Haemorrhagic stroke: caused by a bleeding in or around the brain.-Transient ischaemic attack (TIA): the same as a ischaemic stroke; however, symptoms last for a short amount of time because the blockage of blood supply to the brain is temporary.
*Treatment for stroke* Stroke treatment depends on a number of factors, including cause of stroke and individual risk factors. Below are common medical treatments.-Thrombolysis: clot-busting drug which disperses the clot, usually given within 4.5 h-Thrombectomy: a treatment that physically removes a clot from the brain-Carotid endarterectomy: surgery to unblock fatty deposits from a carotid artery-Antiplatelet drug: help stop clots forming in the blood-Anticoagulant drug: help reduce their risk of developing new blood clots in the future-Blood pressure medicine: help lower high blood pressure-Cholesterol lowering drugs: help lower cholesterol
People who have had a stroke may also may also receive rehabilitation to support their recovery (such as physiotherapy, occupational therapy and psychology) and recommended lifestyle changes to lower stroke risk (such as diet and exercise).*Stroke outcomes* Stroke symptoms and severity are wide ranging. Some people recover quickly; however, many people have long-term needs. The following are common problems experience post-stroke:
-Psychological impact: most common are anxiety and depression, but also anger, frustration, change in emotions-Cognitive impact: includes communication, spatial awareness, memory, concentration, executive function (ability to plan, solve problems and reason about situations) and praxis (ability to carry out skilled physical activities, such as getting dressed)-Physical problems: weakness, paralysis-Communication problems: problems with speaking, understanding reading and writing-Swallowing problems-Visual problems-Bladder and bowel control


## 2. Materials and Methods

### 2.1. Survey Development and Pre-Testing

Survey content was informed by patient partners, vaccine hesitancy literature and the behaviour change model COM-B [24] The survey was reviewed by patient partners, the Stroke Association and the UK/Ireland TIA and minor stroke special interest group. Functionality and usability were tested by the research team and patient partners who piloted completion of the survey and provided feedback, including ease of completion, visual appearance and wording of questions and responses.

The survey comprised multiple choice questions about receipt of the COVID-19 vaccine, perceptions of the vaccine (including safety, access to the appointment, beliefs and social influences, knowledge and understanding) and perspective of COVID-19. There were boxes for free text comments in each section. Demographic information was also collected (Appendix A).

The study was approved by the University of Birmingham Ethical Review Committee on 17 February 2021 (Reference ERN_ 21-0156). Participants provided electronic informed consent and no identifiable information was collected.

### 2.2. Recruitment and Survey Administration

We used an electronic, open survey (i.e., not password protected) hosted by SmartSurvey. Completion was voluntary and no incentives were provided. Participants were eligible if they had experienced a stroke or TIA and were residents in the UK or Ireland. People who experienced a TIA were included as, alongside stroke, this population were considered at greater risk of serious complications of COVID-19 and were eligible for the vaccine the same time as people with stroke [27]. Furthermore, TIA and ischaemic stroke have similar causes and treatments and risk factors [26].

We used convenience sampling with dissemination via social media (Twitter, Facebook); Stroke Association newsletter; and Stroke Association local support services (See Appendix A). In some cases where participants were unable to complete the survey, it was administered by interview from a research nurse or carer.

### 2.3. Context of Vaccine Roll Out

The survey was open between 26 February and 12 April 2021. People who experienced a Stroke/TIA were part of priority group 6 who were eligible to receive the vaccine on 15 February 2021 (i.e., during the time of survey dissemination) [27]. At the time of the survey there were three COVID-19 vaccines approved in the United Kingdom: Moderna vaccine, Oxford/AstraZeneca vaccine and Pfizer/BioNTech vaccine. Survey respondents could have been offered any of these vaccines; however, we did not collect data on which vaccine was offered or received. In March 2021 there was media coverage around the AstraZeneca vaccine and risk of blood clots [28,29].

### 2.4. Patient and Public Involvement and Engagement

The original idea of the survey came from one of our partners, with lived experience of TIA, based on personal experience and from observing discussions on stroke/TIA forums. Our partners with lived experience of stroke/TIA were integral to creating the survey content, testing usability and designing recruitment strategies. In addition, they reviewed the results and provided important lived experience insights to interpret the findings. We continue to work with our lived experience partners to disseminate the findings to people who have experienced a stroke/TIA and key stakeholders.

### 2.5. Data Analysis

Quantitative survey questions were summarised using descriptive statistics. Tests of statistical significance and inferential statistics were not conducted due to the sample size and convivence nature of our sampling. NVivo 12 was used to manage, sort, code and organise free text comments. Qualitative analysis aimed to deepen understanding of perspectives of the vaccine and influences on uptake. Vaccine acceptance and hesitancy are complex behaviours that can that can be potentially influenced by a wide range of factors. Therefore, a model that categorises factors that influence the behavioural decision to accept a vaccine was used as frameworks for a deductive content analysis: the World Health Organisation’s (WHO) 3C’s (Complacency, Convenience and Confidence) model for vaccine hesitancy [6,30]. COM-B was also used in the framework as the survey content was informed by this behavioural model. Text was coded by GT, an experienced qualitative researcher. The final analysis and interpretation were discussed with the research team and patient partners.

## 3. Results

The survey was completed by 377 people who have experienced a stroke and/or TIA. The majority of the sample were White (96.0%: 362/377) and 43.2% (163/377) were male (Table 1). Most of the sample had not experienced COVID-19 (78.8%: 297/377).

### 3.1. Quantitative Survey Questions

#### 3.1.1. Vaccine Uptake

87% (328/377) either had received the first/second vaccine dose or had an appointment booked for vaccine dose (Table 2). The vaccine was declined by 2% (7/377) and 3% (11/377) had been offered the vaccine but not yet taken it up. 8% (30/377) had not been offered the vaccine despite being eligible, of these: 23 were definitely or very likely to accept; 3 were likely to accept; and 4 were unlikely or very unlikely to accept the vaccine.

#### 3.1.2. Perspectives of the Vaccine

##### Side Effects and Safety

Figure 1 summarises survey responses to questions about the vaccine’s safety and side effects. Around a third of respondents strongly agree/agree that they are concerned about:Side effects of the vaccine: 36.0% (131/364)The vaccine increasing their stroke risk: 34.1% (124/364)Safety of the vaccine: 31.6% (115/364)The vaccine affecting blood thinning medication: 29.9% (89/298)How new the vaccine is: 29.1% (106/364).

##### Beliefs and Social Influences

Figure 2 summarises survey responses to questions about the beliefs and social influences. The vast majority of the sample strongly agree/agree that:Having the vaccine is the ‘right thing to do’: 91.8% (328/357)The vaccine will protect against COVID-19: 87.1% (311/357)The vaccine will help reduce spread of COVID-19: 85.1% (304/357).

98.6% (352/357) of respondents strongly agree/agree that they knew other people who have had the vaccine. Very few respondents strongly agree/agree that they have general mistrust of vaccines (6.2%: 22/357) or religious/cultural beliefs affected their decision (3.4%: 12/357).

##### Access to the Vaccine Appointment

Figure 3 summarises survey responses to questions about access to the vaccine appointment. The vast majority of the sample strongly agree/agree that they understood how to get the vaccine (90.8%; 327/360). Most of the sample strongly disagree/disagree that they had difficulty accessing the vaccination appointment (67.8%; 244/360).

##### Knowledge and Understanding

Figure 4 summarises survey responses to questions about knowledge and understanding of the vaccine. Most people strongly agree/agree that they are satisfied with their knowledge and understanding of the vaccine (77.8%; 284/365). Two thirds of respondents strongly agree/agree that the understand where stroke/TIA is on the vaccine priority list (66.8%; 244/365). Nearly half of the sample strongly agree/agree that they searched for vaccine information specifically for stroke/TIA patients (47.5%; 169/356). Only a third (33.2%: 121/365) strongly agree/agree that they were satisfied with the information they found.

The most frequently used sources to get of information about the vaccine were: NHS website (*n* = 167); Stroke Association website (*n* = 151); Google (*n* = 132) and Government website (*n* = 132) (Figure 5).

#### 3.1.3. Perceptions of COVID-19

Figure 6 summarises survey responses to questions about perceptions of COVID-19. 71.5% (254/355) strongly agree/agree that they are likely to pass COVID-19 on to other people if they were to get it. 36.9% (131/355) strongly agree/agree that they are at high risk of getting COVID-19. Over half strongly agree/agree that they will get very sick if they get COVID-19 (57.5%; 204/355) as well as being at greater risk of COVID-19 related complications due to their stroke/TIA (54.4%; 193/355). A third strongly agree/agree that getting COVID-19 would increase their risk of stroke/TIA: 35.8% (127/355).

### 3.2. Qualitative Free Text Comments

#### 3.2.1. Confidence

##### Blood Clots and Stroke Risk (COM-B: Motivation)

Some people expressed serious concerns about blood clots and the vaccine causing a stroke/TIA (64 free text comments).

Media attention around the AstraZeneca vaccine and risk of blood clots caused many people to be anxious and hesitant about the vaccine. In many cases, people had their first vaccine before the media attention and were hesitant to have the second vaccine.


*“Having had dose 1 of AstraZeneca vaccine prior to blood clot issues being reported I am very concerned as to the risks of my second vaccine. Very concerned when I previously had no hesitancy and am actually a vaccinator.”*


Some people experienced a stroke/TIA shortly after having the first vaccine and believed the vaccine caused their stroke/TIA.


*“I had a mini stroke 1 week after receiving first dose. I refuse the second dose of Astrazenica [sic]. I am not being another death statistic.”*



*“17 days after having first vaccine AstraZeneca I had a TIA I am very worried about getting second jab. I’m almost certain it caused me to have a TIA.”*


##### Side Effects and Safety (COM-B: Motivation)

Some people were worried about vaccine side effects after having experienced severe side effects themselves or hearing about potential side effects.


*“The vaccine first dose really put back my recovery by about 3 plus weeks increase in headaches and vertigo. I don’t know whether this means I should or shd [sic] not have second dose.”*


Some concerns were related to comorbidities or medication.


*“My other concerns were I was worried about my other health conditions and medication interfere with the vaccine.”*


Other people’s concerns related to unknown long-term side effects. These concerns were often related to the “newness” of the vaccine.


*“Not sure that it is safe. Wondering if the whole world will be a grand science experiment as there is no long term study on any of the vaccines.”*


##### Vaccine Side Effects

There were 57 unprompted free text comments relating to people’s experience of side effects from the vaccine. A third reported no side effects. Half reported mild or short-lasting side effects, including sore arm, high temperature, cold/flu-like symptoms, headaches, chills, tiredness, sore throat and generally feeling unwell. The remainder reported moderate, severe or long-lasting side effects, including fatigue, severe migraines, headaches, vertigo, feeling dizzy, nausea, muscle weakness and reduced mobility.

##### Mistrust of the Government/Vaccine and Non-Specified Concerns (COM-B: Motivation)

A small minority expressed a mistrust in the government’s response to the virus or the vaccine.


*“I do not trust the goverment [sic] statistics. I don’t see any sign of a pandemic any more than the usual flu outbreaks we get yearly. I feel there is more to this than we are being told. I don’t like the fact we are being controlled and made to feel we have to have a vaccine in my body with pier [sic] pressure. Too many control measures being put on us.”*


Some people had unspecified concerns.


*“I cancelled my first date because I was worried about having it [vaccine].”*


##### Trust/Mistrust in the Vaccine’s Effectiveness (COM-B: Capability)

Some people conveyed their trust in the vaccine’s effectiveness; however, often recognised that it is not a “cure”.


*“The vaccine will reduce the impact of the virus thus preventing admission to hospital. The vaccine is not a cure.”*


Other people were more sceptical about the vaccine’s effectiveness.


*“It is not proven that by getting vaccinated or not is any less likely I wont [sic] get COVID [sic] or pass it on Testing is the best way not vaccination.”*


#### 3.2.2. Complacency

##### Value of the Vaccine: Social and Personal Motivations (COM-B: Opportunity/Motivation)

Some people were motivated to have the vaccine to benefit society and end the pandemic.


*“The vaccine is the only way to get out of this pandemic.”*


For others, motivations were related to protecting family/friends or personal benefits.


*“If you want to see family members or friends you need to have the vaccine to protect them, although I was initially against having the vaccine.”*


A very small minority voiced concerns about potential government-imposed restrictions for people who refuse the vaccine.


*“I think people have the right to take the vaccine or not. I am concerned about the idea of stopping people getting jobs, going to restaurants, travelling etc if they do not have the vaccine.”*


##### Perceived Personal Risk: Knowledge of COVID Risk Related to Stroke/TIA (COM-B: Capability/Motivation)

There was uncertainty about risk of COVID specifically for people who have experienced a stroke or TIA.


*“I have no idea how COVID [sic] impacts on stroke survivors.”*


Some people conveyed concerns about COVID caused stroke/TIA or blood clots.


*“I had COVID [sic] in March 2020 and a tia [sic] in August. I believe COVID [sic] was the cause of my tia [sic]. I do not have a family history of strokes.”*


For some people, concerns about having COIVD were related to other comorbidities.


*“As i [sic] have other health issues aswell [sic] as stroke/TIA I was massively worried with how my body would cope with if i [sic] contracted COVID [sic].”*


#### 3.2.3. Convenience

##### Accessing the Appointment (COM-B: Capability/Opportunity)

Most people had no issues and could access the appointment either independently or with support.


*“My carer arranged the appointment and the travel arrangements, all went smoothly.”*


A minority experienced practical issues booking the appointment.


*“Letter to request I book an appointment on line. I was offered [location A], [location B], [location C], [location D] and some others all miles from my home town. I rang telephone advice line to be told I would have to wait for my doctor or local NHS to contact me for a local appointment. No-one seemed concerned that I was shielding and very vulnerable. It was two weeks later that by telephone I was offered a local appointment.”*


Some people were concerned that they had not been invited for a vaccine appointment yet despite being eligible.


*“I was surprised to only just learn that as a stroke survivor, I am in group 6 for the vaccine rollout. Disappointed to have not been invited for the vaccine by my GP.”*


##### Accessing Information (COM-B: Capability/Opportunity)

Many people did not actively seek information about the vaccine.


*“I didn’t research having the vaccine. I believed that what ever [sic] the effects would be I would ultimately be in a better position for having it than I would of been not having it.”*


Some people were disappointed in the lack of proactive information.


*“I have accessed general information on line but feel my GP surgery should inform its stroke patients with more personalised advice and information for individuals.”*


Some people were frustrated with lack of access to their GP to discuss the vaccine.


*“I contacted my doctor to discuss my jab, but couldn’t get an appointment because the surgery wasn’t really interested in my concerns! No discussion, just a receptionist who said it was safe to have the vaccine! No reassurance for me at all.”*


Some people felt there was a lack of information, in particular personalised information/advice, information on risk of blood clots/stroke and information for younger stroke/TIA patients.


*“I have researched about COVID [sic] and l know that can give blood clots. There is such a lack of advice out there.”*


A very small minority had physical barriers to accessing information.


*“I struggle with information since my strokes hard to take it in an understand it.”*


##### Knowledge about Where Stroke Was on Priority List (COM-B: Capability)

Some people were unclear where stroke was on the vaccine priority list.


*“As far as I am aware people who have had a stroke or TIA were not on a priority list.”*


## 4. Discussion

### 4.1. Principal Findings

Vaccine uptake for the first dose was high: 87% had received the first vaccine or had an appointment booked. Societal and personal benefits may be key motivations for vaccine uptake, in particular perception that having the vaccine is the right thing to do, it will protect against COVID-19, and it will help reduce the spread of COVID-19. Furthermore, the majority were satisfied with their knowledge and understanding of the vaccine. However, some people expressed concerns around the safety of the vaccine (particularly risk of blood clots and stroke) and some were hesitant to have the second vaccine. Most people had no difficulty accessing the vaccine; however, 8% had not been offered the vaccine despite being eligible. There was uncertainty and lack of information about risk of COVID-19 related complications, specifically for people who had a stroke/TIA.

### 4.2. Strengths and Weaknesses

This is the first study to explore perspective of the COVID-19 vaccine from people who have experienced a stroke or TIA in the UK. Although descriptive in nature, our findings enable us to begin to understand behavioural influences on vaccine uptake specific to stroke or TIA. Although it would have been useful to provide correlates of vaccine hesitancy among the targeted population, it was not appropriate to conduct analysis beyond descriptive statistics due to the sample size. A key limitation is that 96% of the sample are White; therefore, perspectives from other ethnic groups may not be represented in our findings. This is particularly important as research has found greater vaccine hesitancy among people from some ethnic minority groups [31]. The survey was only available in English, which hinders participation from non-English language stroke/TIA patients. Furthermore, the survey was electronic and predominantly circulated through social media and email. Therefore, bias may be introduced by digital inaccessibility and the survey under-sampled certain demographic characteristics such as the elderly, particularly the very elderly (>80 years), and lower education status. In the UK, a quarter of all strokes happen to people of working age (<65 years); however, 70% of our sample were <65 years; [25] therefore, employment status in our sample may not reflect the general stroke population. As such, it is unknown if our findings are generalizable to populations under-represented in our survey.

### 4.3. Comparison with Other Studies

Much of the published literature on COVID-19 vaccine hesitancy is conducted pre-/early vaccine roll out and investigates vaccine uptake ‘intent’ [10,11,12,16,32] In contrast, we surveyed vaccine uptake when people who have experienced a stroke/TIA were eligible for the vaccine which enabled us to identify actual, rather than hypothesised, behaviours and perceptions. No other studies have specifically explored the stroke/TIA populations’ perception and uptake of the vaccine.

There was high uptake (87%) of the first COVID-19 vaccine dose in our sample which is consistent with national data from the UK population that found only 7% of adults reported vaccine hesitancy (31 March to 25 April 2021) [33]. However, vaccine uptake in our sample was less than the UK national vaccination rate at the time of the survey: as of 25th April, 93% of the clinically extremely vulnerable population had been vaccinated with at least one dose [34]. We found that around a third of respondents had concerns about side effects and safety of the vaccine. Similarly, qualitative data themes around concerns about the vaccine and risk of blood clots and further stroke. One potential explanation for high vaccine uptake despite safety concerns is that other behavioural influences override safety concerns. Moore et al. found motivations to receive the COVID-19 vaccine among hesitant adopters in the United States were extrinsic motivations, such as protecting community, family, and friends, and intrinsic, such as desire to protect themselves from COVID-19 [35]. Similarly, a UK study found belief it’s the right thing to have the vaccine and desire to protect family members and/or themselves were motivators for people previously unsure whether they would get vaccinated [36]. This aligns with our findings which suggest beliefs and social influences, including personal and societal benefits, may influence uptake. Another potential explanation is that safety concerns emerged after receipt of the first vaccine dose in response. This is supported by our qualitative data where participants expressed hesitancy to have the second vaccine dose following negative media coverage of the AstraZeneca vaccine or because they experienced a stroke/TIA shortly after having the first vaccine and believed the vaccine caused their stroke/TIA. A UK study, conducted in April 2021, found that public demand for AstraZeneca vaccine fell and belief that it causes blood clots increased following blood clot scares [36]. Similar trends for negative perceptions of the AstraZenca vaccine were found in an analysis of Twitter data [37]. Similar to our findings, other studies have identified vaccine safety concerns as important influences on COVID-19 vaccine uptake [14,38,39,40]. Moreover, we found blood clot/stroke risks were particular concerns for people who experienced a stroke or TIA. Concerns about blood clots and stroke side effects may be heightened in the stroke/TIA population because of their previous stroke/TIA experience. Fear of having another stroke is a common form of anxiety post-stroke/TIA [41]. Antivaxxers and conspiracy theories have been discussed as potential influences on vaccine uptake behaviour; however, these were not common influences within our sample [42,43,44]. Similar to our findings, other studies have found risk perceptions (severity of and susceptibility to COVID-19) were significantly associated with vaccine uptake [32].

### 4.4. Implications for Clinicians and Policymakers

The COM-B model of behaviour change maps to “intervention functions” likely to be effective in bringing about change [24]. Therefore, this model can be used as a framework to develop behaviour change interventions and our findings can be used to identify targets for behaviour change to improve vaccine uptake specific to people who experienced a stroke or TIA (Table 3). This is particularly relevant in the context of booster vaccine programmes, declining rates of vaccination and pandemic fatigue [45].

We identified that beliefs and social influences, including societal and personal benefits, were potential key influences on vaccine uptake. However, our survey took place at the early stages of vaccine role out and these motivators may be less prominent as COVID restrictions are eased and the public experiences pandemic fatigue [45]. Therefore, interventions to reinforce beliefs and social influences may be important, including “persuasion”, such as promoting social responsibility and personal gain motivators, and “modelling”, such as positive publicity campaigns of people who have experienced a stroke or TIA having the vaccine.

Importantly, there is a need to increase trust in the safety of the vaccine (confidence). Fear of the vaccine causing blood clots and stroke could be diminished by provision of up-to-date, accessible education/information specifically for people who experienced a stroke or TIA. Evidence suggests that a simple messaging intervention about the safety and efficacy of the COVID vaccine is effective in increasing vaccination intentions [46]. This information should be co-produced with people who have experienced a stroke or TIA and be available through trusted sources, such as government NHS websites [47]. Furthermore, opportunities to discuss individual circumstances with trusted individuals, such as GPs or stroke charity helplines, could be improved. Proactively targeting newly diagnosed people who have experienced a stroke or TIA, in hospital or GP follow-up, provides an opportunity to intervene early to dispel misinformation about the vaccine causing their stroke/TIA and reassurance of safety of second and booster vaccines. Stroke survivors in the community could be targeted through community stroke rehabilitation and stroke groups. Lesson can be learnt from influenza vaccinations in Black, Asian and Minority Ethnic populations, which identified community-based approaches, with local advocacy has potential to counteract misinformation, and concerns regarding side-effects [48].

We identified a lack of understanding that people who have experienced a stroke or TIA have greater risks of complications from contracting COVID-19 (complacency). Therefore, education/information provision to improve knowledge of personal risks related to COVID-19 could increase individuals’ motivation to have the vaccine. A recent study found provision of information on personal benefit reduced vaccine hesitancy to a greater extent for people who are strongly hesitant, compared to provision of information on collective benefits [49].

### 4.5. Unanswered Questions and Future Research

We identified some hesitancy regarding the second vaccine from people who have experienced a stroke or TIA who already received their first dose. Further research is required to understand the proportion of people with stroke or TIA who experience hesitancy resulting in them missing their second or booster vaccine appointments. Understanding the reasons for this hesitancy could inform strategies to improve vaccine uptake. We also identified the need for generation of data about the vaccine and risk of stroke/blood clots specifically for the stroke/TIA population. Ethnic minority, elderly and low education groups are vastly under-represented in our survey; therefore, research which purposively samples these groups is essential to understand influences on vaccine uptake specifically for people who have experienced a stroke or TIA within these minority groups. In the UK, the government is planning a roll out of booster vaccines, starting with the most vulnerable. Further research to understand perspectives and understanding of booster vaccines in the stroke/TIA population is an important continuation of our research to inform strategies to improve vaccine uptake.

## 5. Conclusions

People who experience stroke or TIA are a clinically vulnerable group, at high risk of severe COVID-19. Despite high uptake of the first vaccine, many have legitimate concerns and information needs that should be addressed, in particular regarding risk of blood clots and further strokes, and further interaction with medication. Provision of this information is important to avoid ‘vaccine hesitancy’ in this vulnerable patient group. Our findings can be used to inform targets for behaviour change (such as education, persuasion and modelling) to improve vaccine uptake for people who have experienced a stroke or TIA, in particular increase trust in the vaccine’s safety and improve understanding of the greater risks of complications from contracting COVID-19. This is particularly relevant in the context of booster vaccination programmes.

## Figures and Tables

**Figure 1 ijerph-19-13861-f001:**
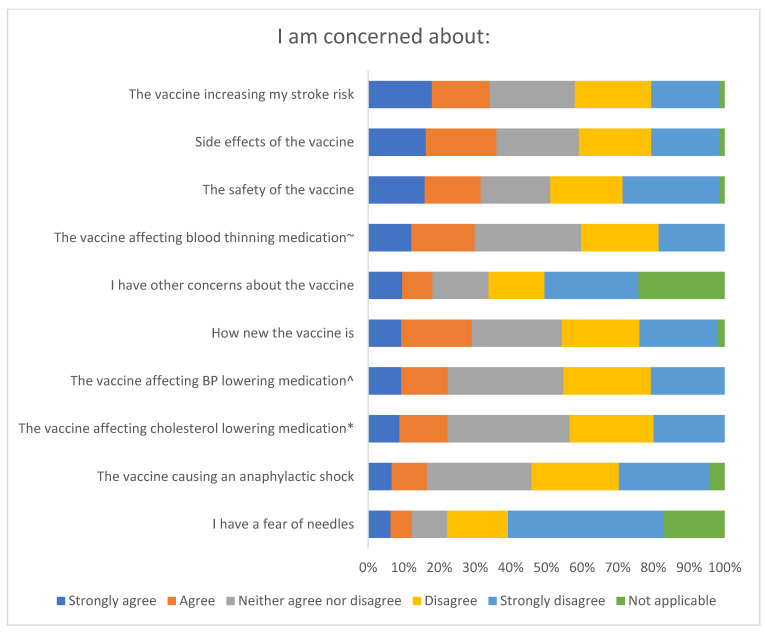
Survey responses to questions about the vaccine’s safety and side effects (*n* = 364, * *n* = 297, ^ *n* = 290, ~ *n* = 298).

**Figure 2 ijerph-19-13861-f002:**
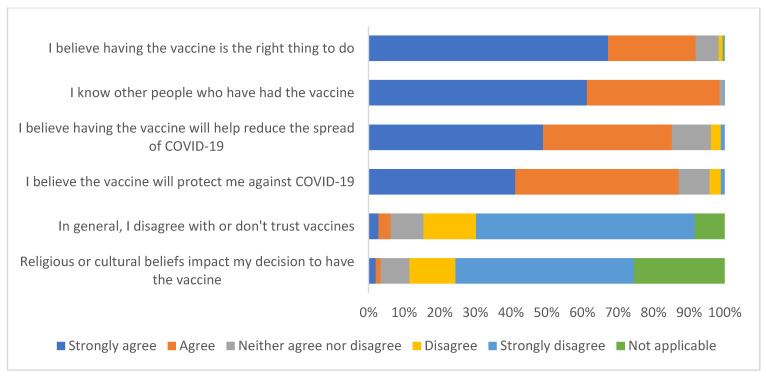
Survey responses to questions about beliefs and social influences (*n* = 357).

**Figure 3 ijerph-19-13861-f003:**
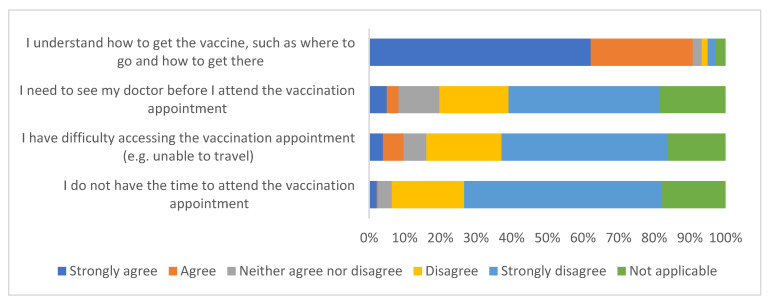
Survey responses to questions about access to the vaccine appointment (*n* = 360).

**Figure 4 ijerph-19-13861-f004:**
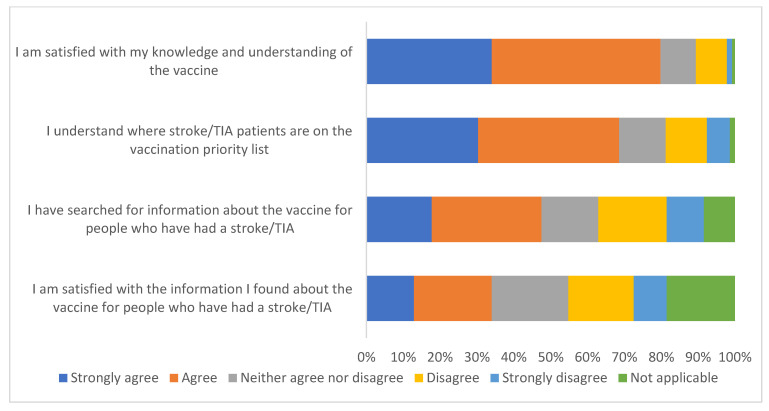
Survey responses to questions about knowledge and understanding of the vaccine (*n* = 365).

**Figure 5 ijerph-19-13861-f005:**
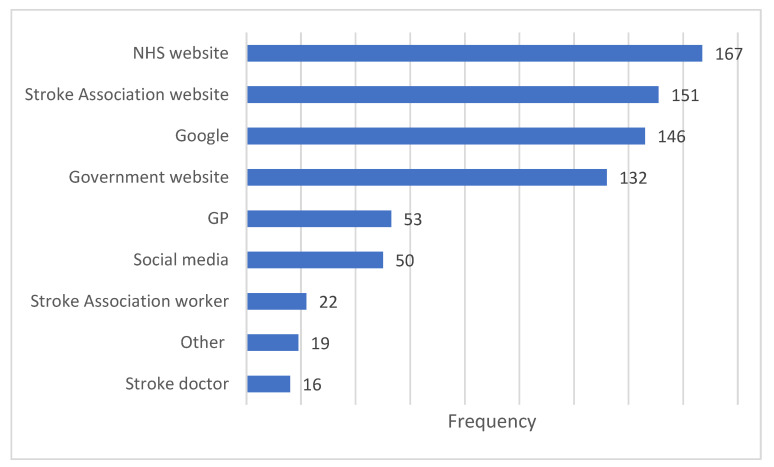
Sources information about the vaccine for people who have had a stroke/TIA.

**Figure 6 ijerph-19-13861-f006:**
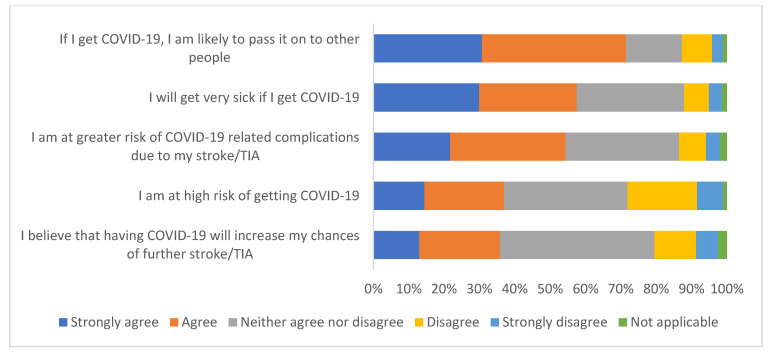
Survey responses to questions about perspectives of COVID-19 (*n* = 355).

**Table 1 ijerph-19-13861-t001:** Demographic characteristics of survey respondents (*n* = 377).

		*n* (%)
Diagnosis	Stroke	253 (67.1)
	TIA	67 (17.8)
	Both	47 (12.5)
	Unsure	10 (2.7)
Gender	Male	163 (43.2)
	Female	213 (56.5)
	Non-binary	1 (0.3)
Age	18–25 years	3 (0.8)
	26–35 years	7 (1.9)
	36–45 years	30 (8.0)
	46–55 years	95 (25.2)
	56–65 years	130 (34.5)
	66–75 years	73 (19.4)
	76–85 years	38 (10.1)
	≥86 years	1 (0.3)
Ethnicity	White	362 (96.0)
	Mixed/multiple ethic groups	3 (0.8)
	Asian/Asian British	5 (1.3)
	Black/African/Caribbean/Black British	5 (1.3)
	Prefer not to say	2 (0.5)
Highest level of education	No formal qualification	36 (9.6)
	GCSE/O-Levels/CSE/Foundation Diploma	100 (26.5)
	Apprenticeship	15 (4.0)
	AS/A-Levels/BTEC/Advanced NVQs	72 (19.1)
	Degree (e.g., BA/BSc)	92 (24.4)
	Higher Degree (e.g., MSc/PhD)	36 (9.6)
	Other	26 (6.9)
Employment status	Employed-full time	80 (21.2)
	Employed-part time	37 (9.8)
	Furloughed-full time	6 (1.6)
	Furloughed-part time	6 (1.6)
	Volunteer	13 (3.4)
	Housewife/househusband	5 (1.3)
	Unemployed	46 (12.2)
	Retired	162 (43.0)
	Student	4 (1.1)
	Other	18 (4.8)
Experienced COVID-19	Yes	44 (11.7)
	No	297 (78.8)
	Unsure	36 (9.6)

**Table 2 ijerph-19-13861-t002:** Vaccine uptake among survey respondents (*n* = 377).

	*n* (%)
Yes-first dose	307 (81.4)
Yes-first and second dose	12 (3.2)
No-I’ve not been offered it yet	30 (8.0)
No-But booked to have vaccine	9 (2.4)
No-I declined the vaccine	7 (1.9)
No-I’ve been offer it but not taken up yet	11 (2.9)
No-Other	1 (0.3)

**Table 3 ijerph-19-13861-t003:** Potential behaviour change targets to improve vaccine uptake in stroke/TIA patients.

3C’s Vaccine Hesitancy Model	COM-B	Intervention Functions	Example
Confidence-vaccine safety	Capability-know the vaccine is safe Motivation-do not have overwhelming fear of the vaccine	Education	Provide clear, concise information on the vaccine and risk of stroke/blood clots, preferably specific to stroke/TIA survivors. Information should be up-to-date and regularly updated as new research emerges. Information should use lay language, be co-produced with patients and be presented visually (e.g., infographics) and using illustrative analogies to contextualise information. Information should be easily available, such as on trusted NHS government websites. Information should be adapted to accommodate accessibility considerations (e.g., visual problems) and stroke-related impairments (e.g., cognitive problems). Empower families/carers to support people with stroke with their information needs.
		Education	Initial information provision in the acute setting is crucial and individual concerns/questions can be discussed. Proactively target newly diagnosed stroke/TIA patients to dispel misinformation about their stroke being related to the vaccine and to promote uptake of the second vaccine. A summary of this information should be included in the discharge letter.
		Education	Educate healthcare providers and vaccinators to provide information to address vaccine safety concerns, particularly regarding blood clot and stroke risk—i.e., more than the top line message ‘the vaccine is safe’. Concerns should be acknowledged and not dismissed. As trusted information sources, healthcare providers should have knowledge of where to access up-to-date, evidence-based information.
		Environmental restructuring	Improve access to personalised advice, support and reassurance from trusted individuals, such as GPs or the Stroke Association helpline.
Complacency-perceived risk of COVID	Capability-understand personal risks Motivation-perceive COVID as health risk	Education	Increase knowledge of the greater risks of complications from contracting COVID-19 for individuals who have had a stroke/TIA. Focus messaging on ‘get the vaccine to reduce your risk’.
Complacency-perceived value of the vaccine	Motivation	Persuasion	Communicate social responsibility and personal benefit to induce positive feelings and stimulate vaccination uptake.
		Modelling	Publicise positive stories of vaccine uptake in stroke/TIA survivors as examples for people to aspire to.

## Data Availability

The datasets used and analysed during the current study are available from the corresponding author on reasonable request.

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
