# Peer review of "Stroke and TIA Survivors’ Perceptions of the COVID-19 Vaccine and Influences on Its Uptake: Cross Sectional Survey"

_ijerph, 2022, doi:10.3390/ijerph192113861_

Round 1
Reviewer 1 Report
Thank you for allowing me to review this article.
It can really be interesting, although the format is inappropriate, it seems more like a student's work than a scientific article.
First of all, box 1, I think it is enough for the expert who reads the article, as well as the rest of the academic explanations.
On the other hand, the tables after the conclusions do not respect the journal's own format.
Lastly, it is not possible to do an article with descriptive statistics, it is necessary at least, inferential statistics.
the paper presented, seems the work of a bad student.
Greetings.
Author Response
We are a team of experienced researchers (3 x PhDs) and clinicians (psychology, occupational therapy and primary care). This survey was supported by a group of stroke survivors and the Stroke Association (UK’s largest stroke charity). Although the methods may appear simplistic, the survey is underpinned by theories of behaviour change and informed by scientific literature and patient and public involvement. The idea for the research question was from a stroke survivor who observed discussions on patient forums and social media about vaccine hesitancy in the stroke population. We responded with rapid methodology that was appropriate for the research question and was developed by experienced researchers. One of the research team is an epidemiologist skilled in advanced statistics. However, inferential statistics were not appropriate for our data due to sample size and rationale underpinning the aims of the survey. We argue the value of descriptive statistics, supplemented by the insightful qualitative analysis, to understanding behaviour and informing policy. To provide an example, qualitative studies were included in a systematic review that, based on findings, developed a framework identified possible factors influencing seasonal influenza vaccination among elderly people: Kan T, Zhang J. Factors influencing seasonal influenza vaccination behaviour among elderly people: a systematic review. Public Health. 2018 Mar;156:67-78. doi: 10.1016/j.puhe.2017.12.007. Epub 2018 Feb 3. PMID: 29408191; PMCID: PMC7111770
We have added to the “data analysis” section (2.5) justification for the descriptive nature of our analyses and role of qualitative analysis to deepen understanding of perspectives of the vaccine and influences on uptake. We hope we have satisfactorily justified why it was not appropriate to conduct inferential statistical analysis of the data given the sample size; however, we have also included this as a limitation in the discussion (section 4.2).
In response to your comment about the journals formatting, we have reformatted the paper to include the figures and tables within the manuscript.
Reviewer 2 Report
Thank you for completing a much needed study of responsiveness to Covid-19 by a high risk population.It would have been useful to provide an even more detailed view of the correlates of vaccine hesitancy among the targeted population.However,as currently presented,the paper offers valuable implications to public health workers for addressing unmet needs of this unique subgroup of the popu6
Author Response
Thank you for your positive feedback. We agree that it would have been useful to provide correlates of vaccine hesitancy among the targeted population; however, it was not appropriate to conduct analysis beyond descriptive statistics due to the sample size, specifically the proportion of people who had not received the vaccine. We have added a sentence reflecting this in the “Strengths and Weaknesses” section (4.2). Nevertheless, we believe our descriptive and qualitative findings are insightful and provide a valuable contribution to understanding behaviour and informing policy.
Reviewer 3 Report
While public health campaigns to protect against covid-19 through large scale community vaccination programs were considered the best path forward to minimize serious and fatal response to covid infection, adoption in some regions, populations, at risk groups, etc. has met with varying degrees of success. Surveys of the public have tried to capture the public opinion of the virus and vaccine in order to understand compliance with public health policies. This study focused on understanding one of the high-risk groups - those with stroke or TIA, and their perceptions and opinions.
The study is timely and useful to understand out current state and plan for future similar situations. The study is commended for working directly with representatives of the target community and various stakeholder organizations. Overall, the quality of writing is high and the methods are thoroughly described. Since the study is complete, there is little/no room for modification of the design, but a small amount of additional data tabulation may be useful. It is unfortunate that the perspectives lack the responses of a more diverse stroke/TIA background. I am sure that the authors and advisors will take this to heart as they plan outreach for future or follow up surveys or health campaigns.
One particular addition would be related to tabulation of Results section 3.1.2 (side effects and safety). These results are reported as ~1/3 of respondents having a concern about 5 different aspects of the vaccine. However, this fails to capture the possibility that potentially 100% of respondents shared serious concerns in general, but deviated by the specific type. Please include a statement that illustrates what total number of all respondents expressed an agreement in general for the 5 specific points. I expect that this may illustrate a much greater proportion of the population expressing a concern, and that is definitely important to communicate.
A second point that may expand on the role of the media reports on AZ vaccine and clot risk and any change in the sentiment of the respondents that may have changed. It would be worth a comment if indeed there appeared to be an influence on the respondents (e.g. more concern expressed after the announcement, other metrics). It would highlight the precise role that reporting/news/media may play in public perception - especially in this particular risk group that are of greater susceptibility of concern relating to their history of stroke/TIA.
Author Response
Thank you for your positive and constructive comments.
In regards to your comment on the “Side effects and safety data” (section 3.1.2), the following statement (3rd bullet point) is less specific and designed to capture general concerns:
- I am concerned about the safety of the vaccine”: 31.6% (115/364) strongly agree/ agree
In regard to your second point about the role of the media reports on the AZ vaccine and clot risk, unfortunately we do not have enough data to quantitatively explore responses pre- and post- media reports. However, our qualitative data suggests that participants’ had concerns specifically related to the news reporting of the AstraZeneca vaccine and risk of blood clots, section 3.2.1:
Media attention around the AstraZeneca vaccine and risk of blood clots caused many people to be anxious and hesitant about the vaccine. In many cases, people had their first vaccine before the media attention and were hesitant to have the second vaccine.
“Having had dose 1 of AstraZeneca vaccine prior to blood clot issues being reported I am very concerned as to the risks of my second vaccine. Very concerned when I previously had no hesitancy and am actually a vaccinator.”